# Current Perspectives on Olfactory Loss in Atypical Parkinsonisms—A Review Article

**DOI:** 10.3390/biomedicines12102257

**Published:** 2024-10-04

**Authors:** Katarzyna Bochniak, Mateusz Soszyński, Natalia Madetko-Alster, Piotr Alster

**Affiliations:** 1Department of Medicine, Faculty of Medicine, Medical University of Warsaw, Żwirki i Wigury 61, 02-091 Warsaw, Poland; s082369@student.wum.edu.pl (K.B.); s082712@student.wum.edu.pl (M.S.); 2Department of Neurology, Medical University of Warsaw, Kondratowicza 8, 03-242 Warsaw, Poland; natalia.madetko@wum.edu.pl

**Keywords:** atypical parkinsonisms, dementia with Lewy bodies, multiple system atrophy, progressive supranuclear palsy, corticobasal degeneration, olfactory loss

## Abstract

Introduction: Atypical parkinsonisms (APs) present various symptoms including motor impairment, cognitive decline, and autonomic dysfunction. Olfactory loss (OL), being a significant non-motor symptom, has emerged as an under-evaluated, yet potentially valuable, feature that might aid in the differential diagnosis of APs. State of the art: The most pronounced OL is usually associated with Dementia with Lewy Bodies (DLB). While the view about the normosmic course of Multiple System Atrophy (MSA) remains unchanged, research indicates that mild OL may occur in a subset of patients with Progressive Supranuclear Palsy (PSP) and Corticobasal Degeneration (CBD). This might be linked to the deposition of abnormal protein aggregates in the central nervous system. Clinical significance: The aim of this review is to discuss the role of OL and its degree and pattern in the pathogenesis and course of APs. Olfactory testing could serve as a non-invasive, quick screening tool to differentiate between APs and project disease progression. Future directions: There is a need for further evaluation of this topic. This may lead to the development of standardized olfactory testing protocols that could be implemented in clinical practice, making differential diagnosis of APs more convenient. Understanding differences in the sense of smell could create an avenue for more targeted therapeutic strategies.

## 1. Introduction

Atypical parkinsonisms (APs) are a collection of rare neurodegenerative diseases that share a complex differential diagnosis process. This group consists of Dementia with Lewy Bodies (DLB), Multiple System Atrophy (MSA), Progressive Supranuclear Palsy (PSP), and Corticobasal Degeneration (CBD). The incidence estimates of atypical parkinsonisms vary between 1 and 7 per 100,000 [1,2]. The data on the prevalence of DLB are ambiguous. DLB cases are estimated to account for about 5% of the total Dementia population, with their prevalence depending on the age of the patient [3,4]. PSP and MSA are found in approximately 6.4 and 4.4 individuals per 100,000, respectively [5]. On the other hand, the prevalence of CBD is estimated to range between 4.9 and 7.3 per 100,000 people [6]. Parkinson’s disease (PD) is commonly misdiagnosed as an AP, especially in its early stages. It is usually marked by bradykinesia with either rigidity or unilateral tremor, accompanied by a clear response to dopaminergic therapy. Furthermore, the presence of a combination of non-motor features, such as autonomic, sleep, and psychiatric dysfunctions, along with hyposmia, should be evident [7]. For this reason, the diagnostic criteria for APs include specific core and supportive features that help differentiate them from the presentation of PD. APs are characterized by a constellation of non-motor and motor symptoms [8]. There is a growing interest in the former, as they play a crucial role in the clinical profile of APs [9,10]. The most common ones such as sleep disturbances, sensory deficits, and autonomic dysfunction often precede motor manifestations and can affect the quality of life [11,12,13].

The diagnostic process of APs presents critical challenges due to their overlapping symptomatology and variability in clinical presentation. The diagnosis is primarily based on a clinical picture, supplemented by imaging. It aims to exclude more common conditions, as there is no definitive diagnostic test for APs [14]. Misdiagnosis remains a significant issue. Limited research dedicated to olfactory loss (OL) among patients with APs negatively affects the development and refinement of diagnostic criteria for each disease entity. The increasing interest in olfactory function in the context of APs is a response to these problems.

It has long been acknowledged that OL seems to be one of the earliest detectable symptoms of neurodegenerative diseases such as PD [15,16,17,18,19]. OL can be divided into anosmia and hyposmia. The severity of the latter can range from mild to profound. It often precedes motor symptoms, which could potentially make OL a valuable marker for early diagnosis. However, a particular disease is not associated with a specific cluster of odors. There is no reproducibility in the correct identification of distinct odors, regardless of the cause of OL [19].

Hypothetically, olfactory testing might improve the prognosis of APs and quality of patients’ life, especially at earlier disease stages. Understanding differences in OL among patients with APs can aid clinicians in reaching a differential diagnosis. Furthermore, analysis of the mechanisms of OL could also enable a more accurate assessment of the underlying pathomechanism.

The aim of this review is to collectively represent the most comprehensive research on the topic of OL during the course of each type of AP.

## 2. Methodology

### 2.1. Search Strategy

A broad search strategy was designed for MEDLINE. It was adapted for further use in the following databases: EMBASE and WEB OF SCIENCE. Additionally, collected data were supplemented using GOOGLE SCHOLAR. These databases were searched from inception to March 2024. In addition, the bibliographies of all relevant systematic reviews were hand-searched.

The search terms combined (Supranuclear Palsy, Progressive [and derivatives] OR Lewy Body Disease [and derivatives] OR Multiple System Atrophy [and derivatives] OR Corticobasal Degeneration [and derivatives]) AND (Olfaction Disorders [and derivatives]). The search phrases consisted of keywords, MeSH terms, and equivalent expressions.

In this review on the olfactory function in patients with specific APs, studies published in the past 15 years were mainly considered. This was caused by limited registries on rare diseases (e.g., CBD), making it difficult to narrow down studies to only the ones that contain patients diagnosed according to the most current criteria.

### 2.2. Study Screening and Selection

The titles and abstracts, as well as the full text of potentially relevant articles were screened by two review authors (KB, MS). The study selection process was documented with a flow diagram (Figure 1).

### 2.3. Study Eligibility Criteria

Original studies, both observational and randomized trials, were analyzed. Papers were excluded if they were an expert opinion, editorial, or conference abstracts. Original papers that included patients with Lewy Body Disease instead of DLB were not considered in this review. Only studies published in English were included.

## 3. State of the Art

APs can be divided into two main categories based on the type of protein deposits in the nervous system: synucleinopathies and tauopathies (Figure 2).

### 3.1. Pathophysiology of Atypical Parkinsonisms

Synucleinopathies are associated with the accumulation of alpha-synuclein, which forms either aggregates or Lewy bodies. Synucleinopathies can be categorized into two disease entities: DLB and MSA. The process of Lewy body formation is not completely understood. Yet they are suspected of disrupting cellular function through various mechanisms. For example, they can participate in the impairment of the ubiquitin–proteasome system and the autophagy–lysosomal pathway [20]. It has been suggested that a prion-like mechanism of spreading misfolded alpha-synuclein contributes to the propagation of Lewy bodies throughout the central nervous system [21].

Tauopathies show the aggregation of abnormal tau, also called microtubule-associated protein tau. The precise role of tau in neurodegenerative processes remains unclear, yet it is known that a loss of tau function precedes degeneration. The sense of smell is discussed in the following tauopathies: PSP and CBD. Phosphorylated tau aggregates form within astrocytes, which leads to their swelling [22]. These aggregates cause destabilization of the microtubules, resulting in the degeneration and death of neurons. Moreover, astrocytes contain the 4-repeat (4R) tau isoform. This accumulation disrupts their normal function, contributing to neurodegenerative processes [23]. Besides their role in microtubules dynamics, tau aggregates are suspected of taking part in other processes that negatively affect cells. They may make DNA more vulnerable to damage, ultimately leading to cell death as well [24] (Figure 3).

According to some studies, abnormal protein aggregates in the olfactory bulb might be the cause of altered smell perception and the reduced ability to detect odors [25,26]. This may indicate that OL is associated with similar pathomechanisms that lead to symptoms such as cognitive decline and motor dysfunction in APs.

### 3.2. Olfactory Dysfunction in Synucleinopathies

#### 3.2.1. Dementia with Lewy Bodies

Olfaction in patients with DLB has already been extensively researched. It has been included in the DLB diagnostic criteria [27]. OL is one of the first disease manifestations. Moreover, as far as 6 years before the disease onset, olfactory disturbances prove to be a highly sensitive and specific marker for various synucleinopathies such as PD and DLB [28,29,30,31].

In DLB, hyposmia manifests itself at an earlier point of the prodromal period of disease when compared to PD. However, patients with DLB have a gentler course of OL progression. Once OL is initially recognized, any further deterioration in olfactory function may be linked mostly to the natural effects of aging [28,29]. Despite that, there seem to be some discrepancies in the literature regarding this relationship between age and olfactory deficits in DLB patients. In a study conducted by Yokogi et al., it is reported that there is no significant correlation between age and OL in patients with DLB [30]. These inconsistencies suggest that factors other than aging may play a role in the progression of OL. Yet distinction between PD and DLB cannot solely rely on olfactory testing. Various studies indicate that the majority of DLB converters may be diagnosed with hyposmia [32,33,34]. It has been observed that patients with DLB exhibit higher alpha-synuclein density found in olfactory brain regions and an overall higher amount of central nervous system alpha-synuclein aggregates compared to PD. Compared to PD, which primarily involves the substantia nigra, olfactory bulbs, brainstem, and limbic regions, DLB shows a significantly higher number of alpha-synuclein aggregates in the cortical regions [35]. The higher alpha-synuclein load in DLB is associated with more rapid disease progression [36]. This difference in alpha-synuclein burden and distribution between DLB and PD may be associated with potential variations in OL. Weakened olfaction is encountered more often in male patients compared with the population of female patients with DLB. This might suggest gender-specific differences in the manifestation and progression of olfactory dysfunction in this neurodegenerative disorder [34].

#### 3.2.2. Multiple System Atrophy

A study on a large cohort confirms that olfaction is preserved in patients with MSA. According to McKay et al., patients initially diagnosed with MSA who exhibited OL eventually developed the DLB phenotype [37]. This study gathered comprehensive clinical cases from patients and their family members, which helps to understand the very early symptoms of MSA and assess the disease stage in every subject. On the other hand, the retrospective nature of the study made the analysis rely markedly on the accuracy of the patient’s and family’s memory, which may not be as reliable. Other studies also show that patients diagnosed with MSA do not develop any signs of OL during the disease [32,38]. Assuming MSA of a parkinsonian type may rarely be manifested by OL; hyposmia in MSA would be substantially less frequent than in PD [39]. The study’s prospective approach allowed for a systematic collection and data follow-up (with a 97.6% rate, which is exceptionally high), enabling the observation of MSA progression over time. Moreover, unexplained anosmia on olfactory testing definitively excludes MSA diagnosis [40].

While University of Pennsylvania Smell Identification Test (UPSIT) scores in patients with MSA seem to be maintained at normosmic levels, Smell Threshold Test (STT) values could indicate mild hyposmia [41]. Furthermore, the Odor Stick Identification Test for the Japanese (OSIT-J) demonstrates a markedly lower sense of smell in patients with PD compared to those with MSA—parkinsonian type (MSA-P), PSP, and control subjects [42]. This phenomenon highlights the need not to rely on a single olfactory test, as it can introduce bias.

In patients with MSA, the temporal and the anterior cingulate cortex show a higher accumulation of alpha-synuclein while the cerebellar white matter and the pons exhibit much lower levels of this protein [43]. This indicates regional variability in the burden of alpha-synuclein within the brains of patients with MSA, which may be linked to the diverse manifestation of OL.

### 3.3. Olfactory Dysfunction in Tauopathies

#### 3.3.1. Progressive Supranuclear Palsy

Discrepancies in the results of various studies may be caused by a lack of differentiation of PSP into disparate subtypes, each exhibiting a distinctive clinical profile, not accompanied by a neuropathological investigation process. These limitations apply to most of the studies analyzed, which negatively impacts the analysis of the collected data and the ability to draw conclusions regarding OL in PSP.

Based on Dutta et al., most patients with PSP present normosmia. Nonetheless, up to 20% of them may show mild OL [44]. This appears to be similar to the clinical picture of MSA. It has been reported that there are no significant differences in the sense of smell between individuals with PSP and those with MSA. Despite this similarity, according to Jia et al., less than 10% of both groups experience hyposmia. This indicates that OL might be present in parkinsonisms other than Parkinson’s disease [45]. However, both populations with PSP and populations with MSA seem to exhibit less-pronounced smell dysfunction compared to most PD patients [46]. Patients with PSP show a wide spectrum of severity of OL ranging from normosmia to levels as profound as those observed in idiopathic Parkinson’s disease. Lower UPSIT scores seem to correlate with greater cumulative Lewy body density, which may be related to deeper cognitive impairment [38,47]. Neuropathology also indicates that a reduced sense of smell might be influenced by Lewy body pathology, which does not meet the criteria for an additional PD or DLB diagnosis [47]. Furthermore, clinical presentation may be affected by tau accumulation in the brainstem and midbrain. Pathological examination of these brain regions reveals the presence of neurofibrillary tangles, composed of hyperphosphorylated tau [48]. Tufted astrocytes and coiled bodies are also visible in tau immunohistochemistry in patients with PSP [49]. Interestingly, OL is not directly mentioned or emphasized in the diagnostic criteria of PSP [50], reflecting perhaps an underappreciation of its clinical significance. Despite this, OL observed in patients with PSP is more severe than in the control population but less pronounced than that reported in the PD group. According to Pavelka et al., every other patient with PSP might experience hyposmia during the disease course [51]. This finding challenges the conventional notion that the entire population of people with PSP is normosmic and prompts a reassessment of the clinical usefulness of olfactory testing, as it may no longer be a discriminant feature favoring Parkinson’s disease over PSP.

#### 3.3.2. Corticobasal Degeneration

Olfaction in patients with CBD is usually preserved. This might be caused by olfactory bulb volume (OBV) and olfactory tract volume remaining unchanged compared to other parkinsonisms where these parameters are often decreased [52]. Normosmia with a decreased DaT score lacks valuable clinical utility in predicting CBD and does not allow for differentiating other APs (PSP, MSA) from idiopathic Parkinson’s Disease (iPD) and DLB [38]. Some studies point out that the population with CBD can present with both significantly impaired and normal olfaction. Research conducted by Luzzi et al. reveals that OL may be indeed present in the population with CBD but with characteristics distinct from those associated with different types of dementia. The authors postulate that since CBD does not affect central nervous system regions that are typically associated with olfaction, the relatively mild problems in odor identification suggest the involvement of an executive component [53]. Current recognition criteria for CBD lack focused insight into the sense of smell of candidates for CBD diagnosis [54]. This could be further explored in future research.

Moreover, some studies investigated OL in Corticobasal Syndrome (CBS) and Frontotemporal Dementia (FTD) rather than focusing on specific pathologies [55,56]. CBS is the most frequently observed clinical manifestation of CBD, yet it is not specific for CBD, as it may also appear in other neurodegenerative diseases. According to Pardini et al., UPSIT scores were notably reduced in the population with CBS. Patients who received a postmortem diagnosis of CBD exhibited anosmia. Furthermore, there were significant correlations between UPSIT scores and gray matter loss in specific brain regions such as the right insula, right midfrontal gyrus, and bilateral inferior frontal gyrus.

The characteristics of OL in each disease have been outlined in Table 1.

## 4. Perspectives and Conclusions

### 4.1. Limitations

This review encounters several limitations that are essential for the interpretation and understanding of the accumulated data. One of them is the inconsistency in sample sizes across different studies. This methodological diversity, while frequently inescapable, remarkably complicates the comparison of study outcomes regarding smell abnormalities. It is especially problematic in the context of rare neurodegenerative disorders such as MSA and CBD. Due to the rarity of these diseases and limited participant numbers, certain groups may be excluded by some researchers, despite their potential relevance in other studies [57]. Because of the long time required to perform olfactory testing on every single patient taking part in the study, some investigators decided to use only hand-picked odors rather than full smell panels [30]. Another problem in interpreting olfactory function in specific disorders belonging to APs is the difficulty in their differential diagnosis, e.g., PSP and CBD. These disease entities exhibit a range of common features, as they both share a similar pathological basis [58,59]. A common issue is the overlapping of the clinical presentation of different diseases, which leads to either misdiagnosis or altered manifestations of the given disease. Some studies lacked neuropathologic confirmation, which is crucial to provide a definite diagnosis of a specific atypical parkinsonism [60].

Ethnicity and environmental impact may markedly influence how odors are perceived and evaluated. This indicates another limitation in this review and olfactory testing in general. It can substantially impact the reliability and universality of olfactory assessments. Certain scents that are easily identifiable in one culture may be unfamiliar in another, leading to variability in identification accuracy [61,62]. The administration of the same olfactory test on patients from different regions or cultures may result in unreliable and ambiguous examination findings [63], which can markedly impact the diagnosis process.

### 4.2. Clinical Significance

OL being an important symptom highlights the usefulness of conducting olfactory tests in the differentiation of parkinsonisms. Alterations in smell often precede cognitive and motor symptoms. A thorough evaluation of olfactory function might provide important prognostic information. Simple olfaction screening tests are affordable and easy to perform, taking only 15 min to administer. They may be especially useful in predicting progression to DLB [31].

Nevertheless, OL is not specific to APs. The degree and pattern of OL might depend on the disease stage and concomitant disorders contributing to OL. This may lead to incomplete management of all symptoms and affect the recognizability of APs. Olfaction can be altered in diseases such as COVID-19 and Alzheimer’s disease (AD) or even during physiological processes, including aging.

Early diagnosis is crucial, as APs can be associated with poor survival (e.g., MSA) [39]. Timely intervention offers the opportunity to implement therapies aimed at not only alleviating symptoms but also delaying disease progression and optimizing patient care. Although olfactory testing should only complement the diagnostic process, it may still prevent diagnostic inaccuracies that eventually result in increased mortality.

Despite a growing understanding of how synucleinopathies progress from their early stages, there remains a need for treatment that could considerably modify the course of these diseases and slow down neurodegenerative processes. There are ongoing trials investigating novel therapies, such as immunotherapy targeting tau, gene therapy, or stem cell therapy [64].

Olfactory testing is relatively inexpensive and quick, and in the case of some diseases it can bring useful diagnostic value. This can help clinicians understand the direction of disease progression and establish the most probable diagnosis. It may also accelerate the implementation of appropriate treatment.

### 4.3. Background and Future Directions in the Analysis of Olfactory Loss

The full pathogenesis of APs is not completely known or confirmed [65,66]. Besides the commonly agreed on influence of alpha-synuclein/tau aggregation on atypical parkinsonism development, neuropathologic studies reveal other potential underlying mechanisms, including inflammatory processes, which may contribute to the onset of the disease [67].

The presence of reactive microglia, causing an inflammatory response in specific parts of the brain, may be potentially linked to OL in neurodegenerative diseases. The GSK3B protein, found in the PSP patients’ central nervous system, is involved in triggering the release of pro-inflammatory mediators secreted by microglia cells. This might sometimes be correlated with hyposmia, which can be observed in the clinical picture of PSP. Similar processes may be noticed in the population with DLB [68].

OL can not only occur as a consequence of neurodegenerative processes but also be the aftermath of various viral infections. The SARS-CoV-2 is one of the most current examples of viruses that often cause OL during an infection [69,70] or even years after it [71]. SARS-CoV-2 infection might be linked with neuroinflammation, which is a common feature of neurodegenerative diseases [72]. The COVID-19 pandemic has led to increased scientific interest in olfactory function and the use of smell tests during the diagnosis of various diseases.

Much more research is needed to better understand the nature of the OL or its absence in every kind of atypical parkinsonism. Researchers should include histopathological examination to confirm a diagnosis made solely on clinical presentation. There is also a need for further exploration of the correlation between inflammation and OL in neurodegenerative disorders. Despite CBD being a rare disease, it should not be overlooked, and further research should aim to refine this disease’s diagnostic criteria. The development of a culturally adapted olfactory test containing various smells and considering all the criteria for odor identification, detection, and discrimination might facilitate the comparison of test results and allow for more definitive conclusions to be drawn.

### 4.4. Differential Diagnosis

For now, there is no diagnostic tool that allows for differentiation between APs, other than postmortem neuropathological examination. Furthermore, some diseases exhibit features similar to those observed in APs, which effectively allows them to mimic these conditions. This lack of specificity makes differential diagnosis challenging not only among the APs themselves but also between other neurodegenerative diseases. This can be seen in PSP, which shares some clinical features with frontotemporal lobar degeneration (FTLD), one of them being mild hyposmia [57].

Another obstacle might be the fact that the clinical picture of APs heavily depends on the stage of disease progression and its subtype [73,74]. As the disease progresses, the clinical picture can undergo significant changes and may exhibit features typical of other neurodegenerative conditions. Similarity between AD and PD with dementia to DLB is commonly reported. However, motor symptoms appear earlier in the population with DLB [75,76]. Clinicians should keep in mind that some symptoms (e.g., bradykinesia, tremor, sleep disturbances, or bladder dysfunction) are non-specific and can occur during the course of multiple diseases.

One potential challenge in using olfactory loss (OL) for differentiating atypical parkinsonian syndromes (APs) is selecting the most suitable method for evaluating OL. As mentioned before, no particular disease is associated with a specific cluster of odors; therefore, there is no “golden standard”. Methods of OL evaluation can be divided into subjective and objective. Subjective methods concentrated on patients’ interpretations of presented odors could be easily misinterpreted due to patients’ influence.

However, subjective tests are simple and fast, which is encouraging in everyday clinical practice. Commonly used subjective tests include, e.g., the Sniffin’ Stick test, the Odorized Markers Test (OMT), or the UPSIT [77].

Objective methods primarily include olfactometry focused on the detection of odorant-evoked neuronal activity in the olfactory pathway [77]. This method objectifies OL severity, excluding potential malingering. However, performing a reliable examination is technically exacting. Therefore, it seems reasonable to perform an easy subjective smell test at first and, in case of doubt, to consider OL objectification with olfactometry.

Olfactory testing may be a useful supporting tool for the differential diagnosis of DLB from other APs. This disease might present with the most profound and the most frequently occurring OL. OL is a new supportive feature for DLB diagnosis according to the current diagnostic criteria [27]. By contrast, the diagnostic criteria for MSA mention unexpected anosmia as one of the exclusion features [40]. Current PSP and CBD diagnostic recommendations do not refer to OL at all [50,54]. The prevalence of OL in APs, as well as inclusion in the diagnostic criteria, are set out in Table 1.

It is reported that patients with idiopathic Rapid eye movement sleep Behavior Disorder (iRBD) who exhibit OL are more likely to develop synucleinopathies compared to normosmic patients [60]. Even though smell assessment in the iRBD population might help identify an individual at a high risk of phenoconversion, it does not distinguish between PD and DLB disease progression [32]. However, intact olfactory function in individuals with iRBD does not necessarily imply a reduced risk of developing synucleinopathy, as they could still be predisposed to MSA conversion [28,60].

The presence of Lewy bodies may be negatively correlated with Brief Smell Identification Test (BSIT) scores [78]. In the prodromal stage of DLB, olfactory dysfunction is already present and correlates with 123I-FP-CIT SPECT scores. Non-motor symptoms in the early stages of DLB are associated with a decrease in DaT binding. Long-standing olfactory dysfunction can signal a more extensive degeneration of the nigrostriatal dopaminergic pathway in the initial phase of DLB [29,38,79]. The combination of isolated abnormal DaT SPECT results and normosmia raises the probability of recognizing an atypical parkinsonism—reaching an approximately 50% chance [38]. This emphasizes the importance of various types of tests in predicting disease progression in patients with iRBD.

The research exploring OBV reveals that, compared to PSP and MSA, mean OBV is reduced in patients with PD, highlighting another potential biomarker for the differentiation of these conditions [44,80].

### 4.5. Conclusions

Patients with DLB present OL more often compared to those who suffer from other APs. The sense of smell in patients with MSA and PSP may also be impaired, despite the different pathogenesis of these diseases. This may suggest the existence of a common underlying mechanism that affects the olfactory system. Abnormal protein aggregates (alpha-synuclein in DLB and MSA; tau in PSP) might alter neural pathways and potentially disrupt cells involved in olfaction, leading to OL. A concluding thought in the context of CBD would be that it is not usually associated with the weakening of smell. The relative rarity of OL in CBD may indicate the involvement of different core mechanisms that would affect olfaction to a much lesser degree. This process is usually accompanied by the deposition of tau.

The current understanding of olfactory disorders in APs is insufficient to establish clear recommendations for distinguishing these conditions based solely on olfactory function. The consideration of olfactory testing can bring many benefits correlated with a greater possibility of correct diagnosis. Exploring the potential entanglement of the olfactory system in APs could open a new avenue for novel treatment strategies.

As of today, OL should only be considered as a supplementary factor in differential diagnosis. More research on this topic is required.

## Figures and Tables

**Figure 1 biomedicines-12-02257-f001:**
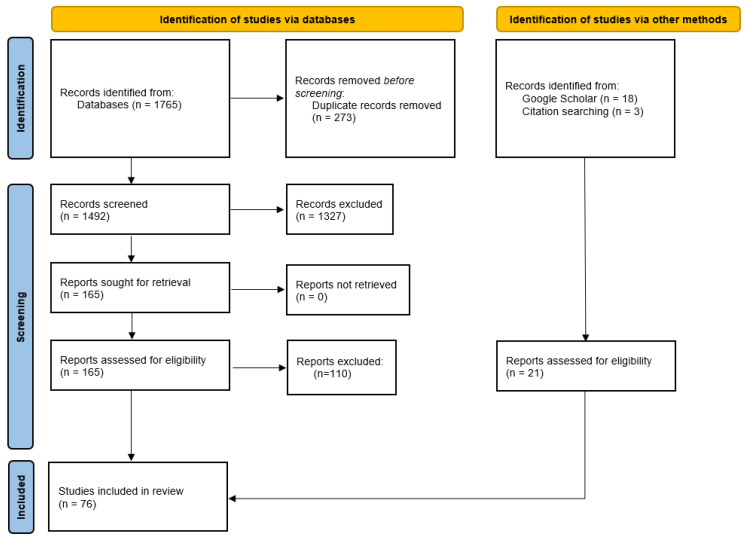
Flow diagram.

**Figure 2 biomedicines-12-02257-f002:**
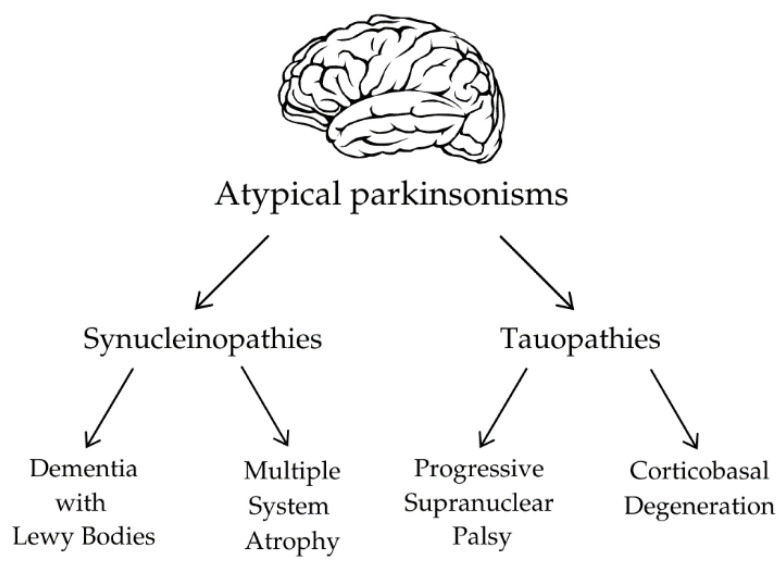
The classification of atypical parkinsonisms.

**Figure 3 biomedicines-12-02257-f003:**
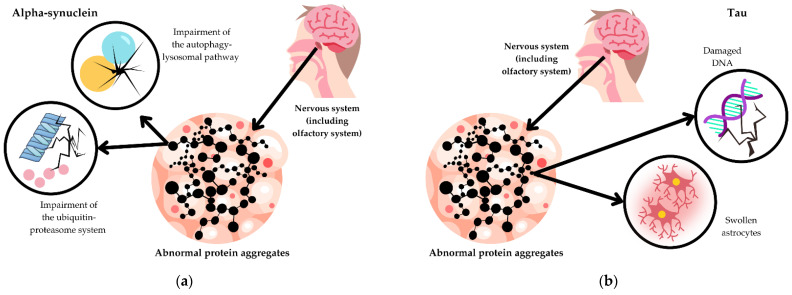
Examples of pathomechanisms underlying atypical parkinsonisms: (**a**) Synucleinopathies; (**b**) tauopathies.

**Table 1 biomedicines-12-02257-t001:** Prevalence and severity of olfactory loss in atypical parkinsonisms *.

Disease		Prevalence of OL	Severity of OL	Reference to OL in Diagnostic Criteria
Synucleinopathies	DLB	+++	profound OL	supportive feature
	MSA	+/−	usually, normosmia	anosmia as an exclusion criterion
Tauopathies	PSP	+	mild OL	-
	CBD	+/−	single studies report hyposmia	-

* Based on the literature in paragraph “state of the art”.

## Data Availability

The original contributions presented in this study are included in the article.

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
