# Peer review of "Current Perspectives on Olfactory Loss in Atypical Parkinsonisms—A Review Article"

_biomedicines, 2024, doi:10.3390/biomedicines12102257_

Round 1

Reviewer 1 Report

Comments and Suggestions for Authors

see attached please.

Author Response

Summary

Thank you very much for taking the time to review this manuscript. We tried to approach all of the issues raised. We hope that the explanations given below will be satisfactory. The modifications in the manuscripts were highlighted in yellow. Point to point responses to your comments and suggestions can be found below.

Comments 1: To improve the clarity and visual attractiveness of your review, consider using pertinent figures and tables. These visual aids can efficiently summarise vital information, demonstrate complicated topics, and make the text more interesting to readers. Incorporating these components will not only improve the overall presentation, but will also attract and hold the attention of a larger audience.

Response 1: Authors appreciate this comment. Authors included three additional figures in the manuscript:

Figure 1 being the Flow diagram (Line 93, Page 3, Methodology)

Figure 2 – The classification of atypical parkinsonisms (Line 102, Page 3, State of the art)

Figure 3 - Examples of pathomechanisms underlying atypical parkinsonisms: synucleinopathies and tauopathies. (Line 124, Page 4, State of the art)

Table 1 (Line 238, Page 7, State of the art), concerning the Prevalence and severity of olfactory loss in atypical parkinsonisms, which was present in the original version of the paper, has been revised to improve readability. Furthermore, it was supplemented with an additional column assessing the severity of olfactory loss.

Comments 2: Authors are recommended to add some general statistics reflecting the incidence or prevalence of disease. How much burden these disorders put on the community to highlight the significance of current study.

Response 2: Authors fully agree with this suggestion. The suggested change was implemented. The estimated incidence of each atypical parkinsonism has been included. The inclusion required the addition of two extra references. “The incidence estimates of the last three are approximately 1 per 100,000 person-years. DLB presents 7 times more often and its appearance is heavily correlated with age [1, 2].” - Page 1, Introduction, Line 36.

Comments 3: Later on this review the authors have mentioned some types of OL such "hyposmia". Therefore, authors are recommended to add brief description of types of OL that takes place in AP for more clarity.

Response 3: The brief description of types of olfactory loss has been incorporated in the following paragraph, according to the suggestion. “OL can be divided into anosmia and hyposmia. The severity of the latter can range from mild to profound.” - Page 2, Introduction, Line 58.

Comments 4: I found the methodology section to be somewhat unclear and perhaps misplaced in the context of this manuscript. Since this is a comprehensive review rather than a systematic review, the inclusion of a detailed methodology section is not typically required. Comprehensive reviews generally synthesize existing knowledge and provide expert insights without the need for a structured methodological approach. I suggest removing or revising this section to better align with the purpose and structure of a comprehensive review, thereby enhancing the readability and focus of your manuscript.

Response 4: Authors appreciate this suggestion. Due to the discrepancy between the reviewers' comments, authors decided not to remove but rather revise the methodology section to make it clearer and more transparent. To support this, a flowchart (Figure 1) was also added. Reviewer 2 suggested the inclusion of databases searched, keywords used, search criteria and a summary using PRISMA flowchart. Bearing in mind that this is a comprehensive review, authors decided to include only vital information that guided the scientific process.

Page 2, Methodology, Line 71.

“2. Methodology

2.1. Search strategy

A broad search strategy was designed for MEDLINE. It was adapted for further use in the following databases: EMBASE, WEB OF SCIENCE. Additionally, collected data was supplemented using GOOGLE SCHOLAR. These databases were searched from inception to March 2024. In addition, the bibliographies of all relevant systematic reviews were handsearched.

The search terms combined (Supranuclear Palsy, Progressive [and derivatives] OR Lewy Body Disease [and derivatives] OR Multiple System Atrophy [and derivatives] OR Corticobasal Degeneration [and derivatives]) AND (Olfaction Disorders [and derivatives]). The search phrases consisted of keywords, MeSH terms and equivalent expressions.

In this review on the olfactory function in patients with specific APs, studies published in the past 15 years were mainly considered. This is caused by limited registries on rare diseases (e.g. CBD), making it difficult to narrow down studies to only the ones that contain patients diagnosed according to the most current criteria.

2.2. Study screening and selection

The titles and abstracts, as well as the full text of potentially relevant articles were independently screened by two review authors (KB, MS). Disagreements were resolved by discussion between the reviewers and, where appropriate, a third reviewer (PA) was consulted. The study selection process was documented with a flow diagram (Figure 1).

2.3. Study eligibility criteria

Original studies, both observational and randomized trials, were analyzed. Papers were excluded if they were an expert opinion, editorial, or conference abstracts. Original papers that included patients with Lewy body diseases instead of DLB, were not considered in this review. Only studies published in English were included.”

Comments 5: At the end of this section, authors are recommended to add at table summarizing the change in olfactory function in different types of AP.

Response 5: According to this suggestion, Table 1 has been modified by adding a new column that provides a clear and concise description of the severity of olfactory loss in each disease entity. We also moved Table 1 from Differential Diagnosis (former Line 319) to the end of the section mentioned in the comment. (Page 7, Line 238, State of the art-> Progressive Supranuclear Palsy)

Comments 6: Again this a review article. Providing discussion section looks irrelevant in such type of articles. What is more relevant is to add a section of conclusions and perspectives here.

Response 6: In response to this comment, we have changed the title of the "Discussion" section to "Perspectives and conclusions" (4. Perspectives and conclusions, Page 7, Line 240). We also included Conclusions as a sub-section (4.5. Conclusions, page 9 Line 376).

4. Additional clarifications

The text was revised 

Reviewer 2 Report

Comments and Suggestions for Authors

Methodology section is incomplete. Needs to discuss search criteria, include keywords search, which databases were search, and include a PRISMA flow chart. 

Comments on the Quality of English Language

Please do not use medical acronyms for titles and list out the word.

Author Response

Initially, we would like to thank the Reviewer for all of their valuable suggestion and comments. We tried to approach all of the issues raised by the Reviewer. We hope that the given below explanations will be satisfactory. Accordingly, point to point responses can be found below.

Comments 1: Methodology section is incomplete. Needs to discuss search criteria, include keywords search, which databases were search, and include a PRISMA flow chart. 

Response 1: Authors appreciate this comment. Due to the discrepancy between the reviewers' comments, authors decided to revise the methodology section to make it clearer and more transparent. To support this, a flowchart (Figure 1) was also added.

Reviewer 1 suggested either a removal of this paragraph or its revision, bearing in mind that this is a comprehensive review. As suggested, authors decided to include only vital information that guided the scientific process, allowing readers to better understand methodological aspects of this paper. Unnecessary formalities related to systematic review type of methodology were therefore omitted.

Page 2, Methodology, Line 71.

“2. Methodology

2.1. Search strategy

A broad search strategy was designed for MEDLINE. It was adapted for further use in the following databases: EMBASE, WEB OF SCIENCE. Additionally, collected data was supplemented using GOOGLE SCHOLAR. These databases were searched from inception to March 2024. In addition, the bibliographies of all relevant systematic reviews were handsearched.

The search terms combined (Supranuclear Palsy, Progressive [and derivatives] OR Lewy Body Disease [and derivatives] OR Multiple System Atrophy [and derivatives] OR Corticobasal Degeneration [and derivatives]) AND (Olfaction Disorders [and derivatives]). The search phrases consisted of keywords, MeSH terms and equivalent expressions.

In this review on the olfactory function in patients with specific APs, studies published in the past 15 years were mainly considered. This is caused by limited registries on rare diseases (e.g. CBD), making it difficult to narrow down studies to only the ones that contain patients diagnosed according to the most current criteria.

2.2. Study screening and selection

The titles and abstracts, as well as the full text of potentially relevant articles were independently screened by two review authors (KB, MS). Disagreements were resolved by discussion between the reviewers and, where appropriate, a third reviewer (PA) was consulted. The study selection process was documented with a flow diagram (Figure 1).

2.3. Study eligibility criteria

Original studies, both observational and randomized trials, were analyzed. Papers were excluded if they were an expert opinion, editorial, or conference abstracts. Original papers that included patients with Lewy body diseases instead of DLB, were not considered in this review. Only studies published in English were included.”

4. Response to Comments on the Quality of English Language

Point 1: Please do not use medical acronyms for titles and list out the word.

Response 1: The suggested change was implemented.

Reviewer 3 Report

Comments and Suggestions for Authors

The review does not present any images; it is recommended to add some images to make the text more fluid. Below are a few examples: Dedoni S, Avdoshina V, Camoglio C, Siddi C, Fratta W, Scherma M, Fadda P. K18- and CAG-hACE2 Transgenic Mouse Models and SARS-CoV-2: Implications for Neurodegeneration Research. Molecules. 2022 Jun 28;27(13):4142. doi: 10.3390/molecules27134142. PMID: 35807384; PMCID: PMC9268291;Farley, Suzanne. "Regulating olfactory receptors." Nature Reviews Neuroscience 5.3 (2004): 171-171.

Line 36: In the introduction, it is recommended to include a brief overview of Parkinson's disease in general, to better understand the other conditions, which are distinctly different.

Line 96: At the beginning of this chapter, I would suggest adding an introductory sentence to help the reader understand the subsequent subchapters, or alternatively, a brief summary table.

Comments on the Quality of English Language

Dear Editor,
Please find below my revisions for this review.

Kind regards 

Ilenia Pinna

Author Response

Initially, we would like to thank the Reviewer for all of their valuable suggestion and comments. We tried to approach all of the issues raised by the Reviewer. We hope that the given below explanations will be satisfactory. Accordingly, point to point responses can be found below.

3. Point-by-point response to Comments and Suggestions for Authors

Comments 1: The review does not present any images; it is recommended to add some images to make the text more fluid. Below are a few examples: Dedoni S, Avdoshina V, Camoglio C, Siddi C, Fratta W, Scherma M, Fadda P. K18- and CAG-hACE2 Transgenic Mouse Models and SARS-CoV-2: Implications for Neurodegeneration Research. Molecules. 2022 Jun 28;27(13):4142. doi: 10.3390/molecules27134142. PMID: 35807384; PMCID: PMC9268291;Farley, Suzanne. "Regulating olfactory receptors." Nature Reviews Neuroscience 5.3 (2004): 171-171.

Response 1: Authors appreciate this comment.. Authors included three additional figures in the manuscript:

Figure 1 being the Flow diagram (Line 93, Page 3, Methodology)

Figure 2 – The classification of atypical parkinsonisms (Line 102, Page 3, State of the art)

Figure 3 - Examples of pathomechanisms underlying atypical parkinsonisms: synucleinopathies and tauopathies. (Line 124, Page 4, State of the art)

Table 1 (Line 238, Page 7, State of the art), concerning the Prevalence and severity of olfactory loss in atypical parkinsonisms, which was present in the original version of the paper, has been revised to improve readability. Furthermore, it was supplemented with an additional column assessing the severity of olfactory loss.

Comments 2: Line 36: In the introduction, it is recommended to include a brief overview of Parkinson's disease in general, to better understand the other conditions, which are distinctly different.

Response 2: The brief overview was added. (Line 38, Page 1, Introduction)

Along with the overview authors decided to include one extra citation to reinforce the claims written.

“Parkinson's disease (PD) is commonly misdiagnosed with AP, especially in the early stages. It is usually marked by bradykinesia with either rigidity or unilateral tremor, accompanied by clear response to dopaminergic therapy. Furthermore, the presence of a combination of non-motor features, such as: autonomic, sleep, psychiatric dysfunction, along with hyposmia, should be evident [3]. For this reason, the diagnostic criteria for APs include specific core and supportive features that help differentiate them from the presentation of PD.”

Comments 3: Line 96: At the beginning of this chapter, I would suggest adding an introductory sentence to help the reader understand the subsequent subchapters, or alternatively, a brief summary table.

Response 3: Having in mind the above comment, authors decided to include a figure (Figure 2, Line 102, State of the art, Page 3) presenting the different types of atypical parkinsonisms, which would be discussed in the later sections. However, the figure got included at the begging of 3 Section. Sections synucleinopathies and tauopathies were renamed to better fit contained content (to Olfactory dysfunction in synucleinopathies/ Olfactory dysfunction in tauopathies accordingly). Furthermore, a new section emerged 3.1. Pathophysiology of atypical parkinsonisms. It contains second and third paragraph of state of the art rearranged, so that the flow and clarity of the text would improve.

4. Response to Comments on the Quality of English Language

Point 1: Dear Editor,
Please find below my revisions for this review.

Kind regards 

Ilenia Pinna
